# Structural Model of the Healthcare Information Security Behavior of Nurses Applying Protection Motivation Theory

**DOI:** 10.3390/ijerph18042084

**Published:** 2021-02-21

**Authors:** EunWon Lee, GyeongAe Seomun

**Affiliations:** 1Department of Nursing, Gwangju University, Gwangju-si 61743, Korea; ewlee@gwangju.ac.kr; 2BK21FOUR R&E Center for Learning Health Systems, College of Nursing, Korea University, Seoul 02841, Korea

**Keywords:** nurse, healthcare information security, protection motivation theory, South Korea

## Abstract

Background: Healthcare information includes sensitive data and, as such, must be secure; however, the risk of healthcare information leakage is increasing. Nurses manage healthcare information in hospitals; however, previous studies have either been conducted on medical workers from various other occupations or have not synthesized various factors. The purpose of this study was to create and prove a model of nurses’ healthcare information security (HIS). The hypothetical model used in this study was constructed on the basis of the protection motivation theory (PMT) proposed by Rogers. Methods: A total of 252 questionnaires scored using a five-point Likert scale were analyzed, incorporating data from nurses who had been working for more than one month in general hospitals with more than 300 beds in South Korea. The survey was conducted over a total of 30 days, from 1 to 30 September 2019. Results: The results showed that coping appraisal significantly influence HIS intentions (estimate = −1.477, *p* < 0.01), whereas HIS intentions significantly influence HIS behavior (estimate = 0.515, *p* < 0.001). A moderating effect on the association between coping appraisal and HIS intentions was found in the group of nurses who had been working for <5 years (estimate = −1.820, *p* < 0.05). Moreover, a moderating effect on the association between HIS intentions and HIS behavior was found in the group of nurses who had been working for <5 years (estimate = 0.600, *p* < 0.001). Conclusion: The results of this study can be used to develop a management plan to strengthen nurses’ HIS behavior and can be used by nursing managers as a basis for developing education programs.

## 1. Introduction

The security of patients’ healthcare information is paramount, as their data may be personalized or discriminated against if disclosed against their permission. Sensitive information related to healthcare history may involve mental illness, sexually transmitted diseases, abortion, and drug abuse [1,2]. Healthcare information may not only be stored in the medical institution at which it was originally created but may also be moved to other institutions with or without a patient’s legal consent [3]. To prevent this, medical institutions are making an effort to secure healthcare information through technical methods, healthcare information management guidelines, and systematic prevention of healthcare information leakage [4].

Information leakage accidents can be divided into technological and human-related incidents, with the latter being more serious [5]. In total, 80% of information leakage stems from insiders, such as former and current employees [6], and internal threats are becoming more common and fatal for organizations [7]. Medical institutions in South Korea are trying to secure healthcare information using centrally controlled technical methods, such as security policies, security education, and computer virus protection software; however, healthcare information security (HIS) still requires attention from healthcare information users [8].

In medical institutions, nurses operate within large workplaces and are in contact with patients for 24 h per day [1]. HIS is important for nurses who directly produce, access, and manage healthcare information in hospitals. Nurses’ HIS behavior can substantially influence the protection of patients’ privacy by preventing healthcare information leakage. The HIS behavior of nurses in contact with patients involves complex interactions of various influencing factors. Therefore, it is important for nurses who directly deal with healthcare information to recognize the importance of HIS [9] and to establish a willingness to practice it [10]. HIS education has been shown to raise nurses’ awareness of the issue and has a positive impact on practice [11,12]. For instance, it has been found that in situations involving increased awareness of the release of healthcare information that could potentially cause irreparable damage to a patient [13], leakage affecting HIS is more likely [5]. In addition, greater self-efficacy leads to greater success of HIS [12,14]. However, physical or environmental obstacles have a negative impact when trying to secure healthcare information [11]. Nurses are hesitate to HIS behavior during the transfer of a patient to another department for examination or when performing nursing services [10]. Nursing services, such as medication or wound dressing, are performed in the patient’s hospital room, and HIS behavior is performed at the nurses’ station. These tasks should be performed in different, separated spaces for HIS behavior [10,14]. In unpredictable and urgent emergencies, such as for those requiring CPR (CardioPulmonary Resuscitation), HIS behavior can be delayed and missed [10]. Additionally, nurses with lengthier careers are statistically more receptive to and proficient in HIS [15]. Nurses who have worked for approximately five years have increased job satisfaction and become proficient in their work [9,10]. In addition, they become better at dealing with crisis situations while conducting their nursing services and often become charge nurses on the ward, meaning that they are in charge of making decisions [9,15]. The Korean Health Industry Promotion Agency conducted a fact-finding survey in which nurses were divided into groups of least five years based on their length of time in the professions [6,10,15]. A prior study identified a number of factors that affect HIS, with some of the identified variables linked to HIS behavior; however, only some studies on HIS behavior have included nurses among a panel of medical institution workers. Furthermore, it is difficult to find studies explaining the HIS behavior of nurses that have compiled all variables related to the process of intent. Nurses’ HIS behavior constitutes complex interactions of various influencing factors at the individual, interpersonal, and environmental levels; thus, the issue should be approached from a theoretical basis to identify the relationships among different influencing factors and to explain and predict nurses’ HIS behavior [4]. Consequently, this study was based on a theoretical model of protection motivation theory (PMT). The aim was to explain the HIS behavior of nurses by applying PMT to the misbehavior of those most involved in the production, maintenance, and security of patients’ healthcare information in medical institutions. To more accurately analyze the correlations among intentions, threats, and responses in an attempt to explain nurses’ HIS behavior, we present a predictive model. As a result, basic data are provided that can be used for the development of education programs that strengthen the HIS behavior of nurses, whereas nursing managers and policymakers can receive assistance in coming up with countermeasures against HIS.

## 2. Research Hypotheses

### 2.1. Theoretical Foundation

#### Protection Motivation Theory

Protection motivation theory explains how individuals are exposed to threats and how their attitudes and behavior consequently change [16,17]. Rogers [17] stated that when an individual is exposed to a threatening situation, protection motivation is formed through threat and coping assessments, thereby leading to a particular action being taken. In PMT, subfactors related to threat appraisal include intrinsic rewards, extrinsic rewards, perceived severity, and perceived vulnerability, while subfactors related to coping appraisal include response efficacy, self-efficacy, and response costs. Rogers and Price-Dunn [17] proposed that PMT is applicable if any threat, including those associated with health promotion, disease prevention, injury prevention, and the protection of others, can be effectively prevented at a personal level. Thus, PMT was applied to research information security behavior related to complying with security policies [16,17], as well as to the research of disease prevention, and this has been extended to various cases where protection behavior is required in the face of accident prevention or external threats [17]. Therefore, this study used PMT as a theoretical framework.

### 2.2. Anticipated Outcomes

PMT considers threat appraisal to be the assessment of threats upon an individual’s exposure [16,17]. Threat appraisal is a person’s assessment of the degree of threat when exposed to threatening events [17]. In this study, threat appraisal was defined as the extent to which nurses are exposed to HIS threats. Additionally, we considered the intention to be the adoption of actions to protect oneself from threats [17]. Subfactors related to threat appraisal include intrinsic rewards, extrinsic rewards, severity, and vulnerability, according to the results of prior studies [5,12,18,19,20,21,22,23,24], leading to the following hypothesis:

**Hypothesis** **1** **(H1).**
*The threat appraisal of HIS by nurses will affect their intentions.*


Protection motivation theory explains coping appraisal as the assessment of an individual’s ability to cope with losses upon exposure to threats [16,17]. In this study, coping appraisal was defined as the extent to which nurses are able to cope with losses arising from HIS threats. Subfactors related to coping appraisal include response efficacy, self-efficacy, and response costs, according to the results of prior studies [18,20,21,24,25,26,27], leading to the following hypothesis:

**Hypothesis** **2** **(H2).**
*The coping appraisal of HIS by nurses will affect their intentions.*


Protection motivation theory explains how individuals are exposed to threats, as well as the subsequent changes in their attitudes and behavior [16,17]. Behavior is an act of protecting oneself [17]. Threat appraisal and coping appraisal result in the protection of motivation and changes in behavior through the process of intention, according to the results of prior studies linking these factors to HIS behavior [12,22,23,24,28,29,30], leading to the following hypothesis:

**Hypothesis** **3** **(H3).**
*The threat appraisal of HIS by nurses will affect their behavior.*


In this study, protection motivation was defined as the extent to which nurses are able to cope with losses arising from HIS threats, according to the results of prior studies [5,9,11,12,14,21,24,25,26,27,31], leading to the following hypothesis:

**Hypothesis** **4** **(H4).**
*The coping appraisal of HIS by nurses will affect their behavior.*


When a subject is exposed to a threatening message, they assess the threat to determine whether a certain coping approach will work. According to the findings of prior research [12,14,19], this results in a change in behavior through the process of intent, thereby creating protection motivation, leading to the following hypothesis:

**Hypothesis** **5** **(H5).**
*Nurses’ HIS intentions will affect their behavior.*


Based on the results identified in the literature review, a theoretical model was developed to incorporate the abovementioned hypotheses, as shown in Figure 1.

## 3. Research Methodology

### 3.1. Ethics Statement

This study conducted a survey with the approval of the Institutional Review Board (IRB) of Korea University (IRB No. KUIRB-2019-0200-01). After obtaining approval from the nursing department, a public notice for the recruitment of eligible persons was posted. The researchers ensured that every participant understood the research objectives and procedures before acquiring informed consent for their involvement. It was explained that the anonymity of the subjects was guaranteed and that the findings would not be used for any purpose other than research in the future. Nurses participated in the survey after being informed that it would take approximately 20 min. After explanation, all of them signed the consent form. The nurses who participated in the survey were given a small travel wash kit to thank them.

### 3.2. Participants and Procedures

The inclusion criteria included nurses who had worked at general hospitals with 300 beds or more in Seoul and Gyeonggi-do, the largest area near Seoul, South Korea, for more than one month, who understood the purpose and methods of this research, and who agreed to participate in the survey. Nurses who did not use healthcare information systems, such as the Ordering Communication System (OCS), the Picture Archiving Communication System (PACS), and Electronic Medical Record (EMR), were excluded. Hospitals with more than 300 beds operate more medical departments, so OCS, EMR, and PACS are equipped to process healthcare information electronically, making it appropriate to investigate nurses’ HIS. Nurses with less than one month of work experience were excluded, as they do not yet perform HIS behavior independently because they perform all duties under supervision in one’ ward during the training period. After obtaining approval from the nursing departments of the general hospitals, a notice of recruitment was posted. Descriptions of the objectives and procedures involved in this study were available in a designated location for nurses who wished and agreed to participate. The survey was conducted for a total of 30 days, from 1 to 30 September 2019. A total of 252 questionnaires were analyzed.

### 3.3. Measures

To measure the concept of nurses’ HIS behavior, we used the PMT variables proposed by Rogers and Price-Dunn [17]. Threat appraisal and coping appraisal were set as independent variables, and HIS intention and HIS behavior were set as dependent variables [16,17]. Intrinsic rewards, extrinsic rewards, severity, and vulnerability were selected as subfactors of threat appraisal [17,25]. Response efficacy, self-efficacy, and response costs were selected as subfactors of coping appraisal [17,27]. Career length was set as a moderate variable [10].

In this study, HIS intention was measured by the tools used by Kim [32]. The measurement tool involved five questions, each answered on a five-point Likert scale ranging from 1 (“strongly disagree”) to 5 (“strongly agree”): The higher the score, the higher the intention for HIS. In a study by Kim [32], the value of Cronbach’s α was 0.890, and it was 0.856 in this study.

In this study, HIS behavior was measured by the tools used by Kim [32]. The measurement tool included 10 questions, each answered on a five-point Likert scale ranging from 1 (“strongly disagree”) to 5 (“strongly agree”): The higher the score, the more strongly the HIS behavior was practiced. In a study by Kim [32], the value of Cronbach’s α was 0.765, and it was 0.869 in this study.

Intrinsic rewards describe one’s satisfaction or sense of accomplishment [17,25]. In this study, intrinsic rewards were measured by the tools used by Kim et al. [33]. The measurement tool involved three questions, each answered on a five-point Likert scale ranging from 1 (“strongly disagree”) to 5 (“strongly agree”): The higher the score, the higher the intrinsic rewards for HIS. In a study by Kim et al. [33], the value of Cronbach’s α was 0.954, and it was 0.947 in this study.

Extrinsic rewards include social consensus, peer influence, and education [17,25]. In this study, extrinsic rewards were measured by the tools used by Kim et al. [33]. The measurement tool involved four questions, each answered on a five-point Likert scale ranging from 1 (“strongly disagree”) to 5 (“strongly agree”): The higher the score, the higher the extrinsic rewards for HIS. In a study by Kim et al. [33], the value of Cronbach’s α was 0.931, and it was 0.874 in this study.

Severity is the extent to which a hazard is fatal if it occurs [17,25]. In this study, severity was measured by the tools used by Jung [34]. The measurement tool involved six questions, each answered on a five-point Likert scale ranging from 1 (“strongly disagree”) to 5 (“strongly agree”): The higher the score, the higher the severity related to HIS. In a study by Jung [34], the value of CSRI (Composite Scale Reliability Index) was 0.942, and the value of Cronbach’s α was 0.896 in this study.

Vulnerability is the possibility that a hazard will actually occur [17,27]. In this study, vulnerability was measured by the tools used by Jung [34]. The measurement tool involved six questions, each answered on a five-point Likert scale ranging from 1 (“strongly disagree”) to 5 (“strongly agree”): The higher the score, the higher the vulnerability related to HIS. In a study by in Jung [34], the value of CSRI was 0.945, and Cronbach’s α was 0.809 in this study.

Response efficacy is whether the proposed policy has the effect of preventing the hazard policy [17,25,27]. In this study, response efficacy was measured by the tools used by Son [35]. The measurement tool involved four questions, each answered on a five-point Likert scale ranging from 1 (“strongly disagree”) to 5 (“strongly agree”): The higher the score, the higher the response efficacy related to HIS. In a study by Son [35], the value of Cronbach’s α was 0.877, and it was 0.758 in this study.

Self-efficacy involves the self-assessment of whether one can carry out the proposed policy [17,25]. In this study, self-efficacy was measured by the tools used by Son [35]. The measurement tool involved four questions, each answered on a five-point Likert scale ranging from 1 (“strongly disagree”) to 5 (“strongly agree”): The higher the score, the higher the self-efficacy related to HIS. In a study by Son [35], the value of Cronbach’s α was 0.929, and it was 0.880 in this study.

Response costs represent obstacles, such as time and hassle, that make it difficult to actually conduct a process [17,27]. In this study, response costs were measured by the tools used by Son [35]. The measurement tool involved three questions, each answered on a five-point Likert scale ranging from 1 (“strongly disagree”) to 5 (“strongly agree”): The higher the score, the higher the response costs related to HIS. In a study by Son [35], the value of Cronbach’s α was 0.795, and it was 0.772 in this study.

Nurses’ careers were classified as less than or more than five years [9,10,15].

### 3.4. Data Analysis

The collected data were analyzed using SPSS Version 22 (64 bit Korean Version IBM, New York, NY, USA) and AMOS 21(IBM) (IBM, New York, NY, USA). An exploratory factor analysis was conducted to analyze the validity of the measuring tool. In constructing the structural equation model, the validity of the measurement tool was finally tested using a confirmatory factor analysis. The validity of the hypothetical model was analyzed using the chi-squared value, the goodness of fit index (GFI), the adjusted goodness of fit index (AGFI), the rootmeansquare error or approximation (RMSEA), the rootmeansquared residual (RMR), and the comparative fit index (CFI). Moderating effects were confirmed according to the nurses’ hospital-based careers.

## 4. Results

### 4.1. Demographic Characteristics

A total of 260 questionnaires were distributed to the nurses who voluntarily participated in the study. In total, 259 were received, providing a response rate of 99%. After removing questionnaires containing incomplete or missing responses to more than 30% of the questions, the number of remaining valid questionnaires was 252. In terms of demographics, female respondents accounted for 96.4% of the total. Most respondents were aged between 25 and 29 years, accounting for 47.6% of the total. The majority of the respondents were attending charge nurses (63.9%) who had been working for one to four years (36.5%). The participants’ complete demographic data are detailed in Table 1.

### 4.2. Measurement Model

First, to analyze the general characteristics of the subjects and the validity of the tool used, Cronbach’s α internal reliability coefficient values for the exploratory factor analysis (EFA) and the reliability test were analyzed by Pearson’s correlation. Second, using a confirmatory factor analysis (CFA), the measurement model was tested in terms of its content, convergent, and discriminant validity. This study reviewed and adapted the constructs and measurement items. Cronbach’s α, composite reliability (CR), and the average variance extracted (AVE) were examined to evaluate the convergent validity.

In this study, we determined the convergent validity of the selected variables. The mean and standard deviation of each variable were obtained. The obtained Cronbach’s α and CR values were higher than the acceptable threshold of 0.7, and the AVE was higher than the acceptable threshold of 0.5. Thus, the presence of convergent validity was supported. Meanwhile, interconstruct correlation coefficients were tested to measure the discriminant validity. The square roots of the AVEs for each construct were higher than the other values in the corresponding columns and rows, thus verifying the presence of discriminant validity. As a result, the measurement model used in this study was validated. The convergent validity results are detailed in Table 2, and the discriminant validity results are detailed in Table 3.

### 4.3. Initial Hypothetical Model

In this study, we determined the discriminant validity of the selected variables. Discriminant validity indicates different latent variables. Having low correlations between latent variables indicate discriminant validity. This study model consisted of HIS threat appraisal, HIS coping appraisal, HIS intentions, and HIS behavior. The factors derived from the EFA in the study model were used to demonstrate the validity of the factor variables through CFA. To verify the validity of the concept, GFI, AGFI, CFI, RMR, NFI, RMSEA, chi-squared, and *p*-values were analyzed. The results of the hypothetical model test were as follows: χ^2^ = 131.035 (df = 23, *p* < 0.001), GFI = 0.896, AGFI = 0.458, CFI = 0.810, RMR = 0.055, NFI = 0.783, and RMSEA = 0.137. The initial hypothetical model is shown in Figure 2.

### 4.4. Revised Hypothetical Model

During the analysis of the initial hypothetical model, the fit index could not be determined; thus, the initial hypothetical model was revised using modification indices (MIs; Lagrange multiplier tests). The covariance among variables was set according to the MIs, resulting in the chi-squared value decreasing and the fit improving. The results of the revised hypothetical model test were as follows: χ^2^ = 111.445 (df = 21, *p* < 0.001), GFI = 0.916, AGFI = 0.819, CFI = 0.84, RMR = 0.050, NFI = 0.815, and RMSEA = 0.131. Thus, it was confirmed that the influence of the revised model was improved with respect to the initial hypothetical model. The revised hypothetical model is shown in Figure 3.

The results of the hypothetical test conducted with the revised study model are described below.

**Hypothesis** **1** **(H1).**
*The threat appraisal of HIS by nurses did not affect their intentions, so H1 was rejected.*


**Hypothesis** **2** **(H2).**
*The coping appraisal of HIS by nurses affected their intentions, so H2 was accepted.*


**Hypothesis** **3** **(H3).**
*The threat appraisal of HIS by nurses did not affect their behavior, so H3 was rejected.*


**Hypothesis** **4** **(H4).**
*The coping appraisal of HIS by nurses did not affect their behavior, so H4 was rejected.*


**Hypothesis** **5** **(H5).**
*Nurses’ HIS intentions affected their behavior, so H5 was accepted.*


## 5. Moderating Effect of Length of Nursing Career in a Hospital

The greater a nurse’s experience, the more significant an impact HIS has on behavior [15]. In particular, nurses who have worked for more than five years have been found to be statistically significantly more likely to carry out HIS behavior than nurses who have worked for less than five years [9]. These previous studies showed that HIS intentions and HIS behavior vary significantly depending on career length (shorter or longer than five years) [14]. Additional verification was required to explain the impact of career length on HIS threat appraisal, HIS coping appraisal, HIS intentions, and HIS behavior. According to the results of prior studies showing such a relationship, the following hypotheses were proposed to identify the moderating effect of having more than five years of experience:

**Hypothesis** **6** **(H6).**
*The relationship between a nurse’s HIS threat appraisal and their intentions will receive a moderating effect from the length of their career in a hospital.*


**Hypothesis** **7** **(H7).**
*The relationship between a nurse’s HIS coping appraisal and their intentions will receive a moderating effect from the length of their career in a hospital.*


**Hypothesis** **8** **(H8).**
*The relationship between a nurse’s HIS threat appraisal and their behavior will receive a moderating effect from the length of their careers in a hospital.*


**Hypothesis** **9** **(H9).**
*The relationship between a nurse’s HIS coping appraisal and their behavior will receive a moderating effect from the length of their career in a hospital.*


**Hypothesis** **10** **(H10).**
*The relationship between a nurse’s HIS intentions and their behavior will receive a moderating effect from the length of their career in a hospital.*


In order to verify the above mentioned moderating effects according to the nurses’ hospital careers, the previously proposed structure model was rearranged by categorizing the respondents according to the length of their career (<5 years vs. ≥5 years). The results of this new model were as follows: χ^2^ = 17.757 (df = 5, *p* < 0.01), NFI = 0.027, IFI = 0.029, RFI = 0.005, and TLI = 0.005. Therefore, the moderating effect of the nurses’ length of career working in a hospital was confirmed, and this is detailed in Table 4.

## 6. Discussion

This study hypothesized that the threat and coping appraisals of nurses related to HIS influence their intentions and behavior. In addition, we checked whether there were any moderating effects due to the nurses’ career lengths. We identified statistically significant effects for two of the five hypotheses related to principal factors and two of the five hypotheses related to moderating effects, leading to their acceptance.

Threat appraisal, which included intrinsic rewards, extrinsic rewards, severity, and vulnerability as subfactors, was found to have no effect on the nurses’ HIS intentions and behavior. The results of the studies by Kim et al. [33] and Holen et al. [20], who reported more beneficial HIS intentions and behavior effects, are in contrast to the results of this study. This suggests that nurses are aware of the importance of HIS behavior and the associated risks to patients, and do not simply seek intrinsic rewards such as satisfaction and achievement when exhibiting HIS behavior. As for the results of our study, personal satisfaction and a sense of achievement can provide motivation when carrying out nursing work; however, nurses should not perform their work solely to receive intrinsic rewards, but rather, through expertise and ethical awareness, according to the results of [33]. Box et al. [14] conducted a systematic review of the factors affecting HIS behavior in healthcare, confirming that extrinsic rewards such as peer influence do not affect HIS intentions or behavior. They found that the value and importance of HIS behavior should be understood and that procedures such as the enforcement of regulations should be clearly notified in an attempt to achieve HIS behavior. It is necessary to actively intervene to improve the HIS behavior of nurses by providing education through simulated situations. Karjalaninen et al. [36] suggested that, since members of medical institutions are always at risk of HIS accidents when on duty, training should be conducted such that these accidents can always be recognized. Today, not only medical institutions but also major corporations and government organizations are facing the reality that they may be hacked [37], so information security policies should be implemented for each institution to ensure information security. As for the results of our study, medical institutions can also strengthen their HIS behavior by providing empirical guidelines and ensuring that their institution workers remain familiar with them by checking relevant details within OCS, PACS, and EMR after work

Coping appraisal, which includes response efficacy, self-efficacy, and response costs as subfactors, was found to have an effect on nurses’ HIS intentions, but not on their behavior. Research by Kajtazi et al. [24] showed that response efficacy affects HIS information (HISI) and HIS behavior (HISB). It is believed that nurses currently comply with the HIS recommendations and enforcement rules proposed by medical institutions and that compliance can prevent the leakage of healthcare information. Kajtazi et al. [24] also showed that greater self-efficacy related to compliance leads to improved HIS intentions, whereas Shahri et al. [11] showed a similar correlation with improved HIS behavior. Our results indicate that it is necessary for interim nurse managers and supervisors to apply achievement-oriented leadership rather than directed leadership [9], thereby exhibiting confidence in nurses who voluntarily perform HIS behavior for its inherent purpose.

Nurses’ HIS intentions were found to influence their behavior. It is important to study HIS intentions to identify factors potentially leading to changes in behavior [23]. This study examined the variables affecting nurses’ HIS behavior, confirming that greater HIS intentions are correlated with stricter compliance with HIS behavior. HIS intentions act as mediating factors for the promotion of HIS behavior by preventing leakage in medical institutions. Nurses are a very important group of healthcare professionals in medical institutions, and they must deal with disaster management [38], including HIS leakage. Nurses who have communication skills and knowledge on information utilization will be able to cope with emergency situations such as HIS leakage [38,39]. Major corporations and government organizations are actively protecting information through the use of new methods such as the FORGE system or MSN modeling [37,40]. Medical institutions may also develop new technologies for HIS or modify the previously mentioned methods to suit medical institutions. Our results indicate that nurses with appropriate HIS intentions practice the enforcement rules proposed by medical institutions to prevent the leakage of medical information, which then directly affects their behavior, preventing the patient disadvantages and discrimination associated with leakage.

We found that career length had moderating effects on HIS threat appraisal, HIS coping appraisal, HIS intentions, and HIS behavior. This result shows that the greater the work experience, the more HIS behavior is displayed. This shows an association between practical experience and proficiency in complying with the rules proposed by medical institutions. It is also believed that, as the number of training sessions increases, recognition of the need for, and effectiveness of, using security systems increases. This result shows that an increased career length is associated with repeated educated and thus improved awareness of the need for, and effect of, using a security system. Thus, educational programs should be enhanced by reflecting content classified according to work experience to improve HIS behavior. In particular, it is suggested that nurses with less than five years of work experience are targeted.

Generalization to all age groups has its limitations, but the data collected in this study can be best applied to nurses aged 25–29 years. This age group generally includes charge nurses, so the data can be used as basic information when creating policies for nurses in medical institutions. The other limitation of this study is that the only variable in the PMT was the description of nurses’ HIS behavior. It is necessary to measure HIS behavior according to various types of theories. However, the use of threat appraisal, coping appraisal, intention, and behavior in PMT is appropriate for explaining HIS. Prior studies only examined HIS intentions or behavior, but this study verified its methods using SEM to explain HIS intentions and behavior at the same time. Furthermore, prior studies only described HIS as a variable of PMT [6,10,15,28,29,30], but this study described HIS factors that interact with various variables in PMT. Our results indicate that nurses who have worked for more than five years become proficient in their work and have improved job satisfaction are more likely to exhibit HIS behavior than nurses who have worked for less than five years. Therefore, this study presented a suitable model for explaining and predicting the HIS behavior of nurses.

## 7. Conclusions

As the possibility of healthcare information leakage increases, medical institutions are striving for information security. This study was conducted to explain the HIS behavior of nurses. The HIS behavior of nurses is very important because there are large numbers of nurses in medical institutions, and nurses are with patients 24 h a day. Prior studies only described the HIS intentions or behavior of nurses as parts of a variable, but this study explains the HIS intentions and behavior of nurses simultaneously by applying PMT. In this study, coping appraisal was found to influence HIS intentions, and HIS intentions were found to influence HIS behavior. Therefore, the results of this study indicate that in order to increase the adoption of HIS behavior, HIS intentions need to be improved, and for this, coping appraisal needs to be improved.

## Figures and Tables

**Figure 1 ijerph-18-02084-f001:**
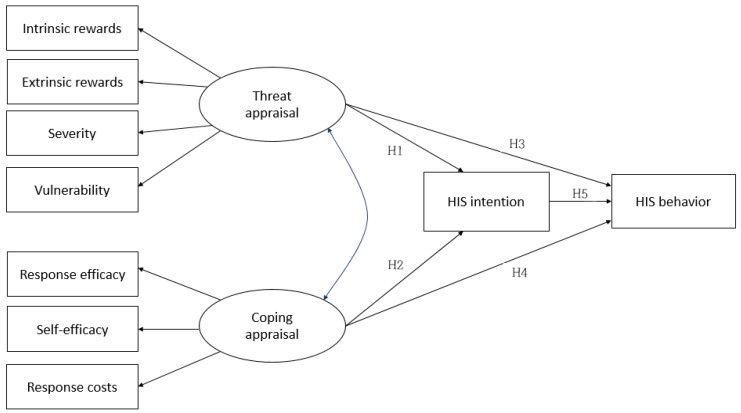
Hypothetical model. HIS, healthcare information security. H1: The threat appraisal of HIS by nurses will affect their intentions. H2: The coping appraisal of HIS by nurses will affect their intentions. H3: The threat appraisal of HIS by nurses will affect their behavior. H4: The coping appraisal of HIS by nurses will affect their behavior. H5: Nurses’ HIS intentions will affect their behavior.

**Figure 2 ijerph-18-02084-f002:**
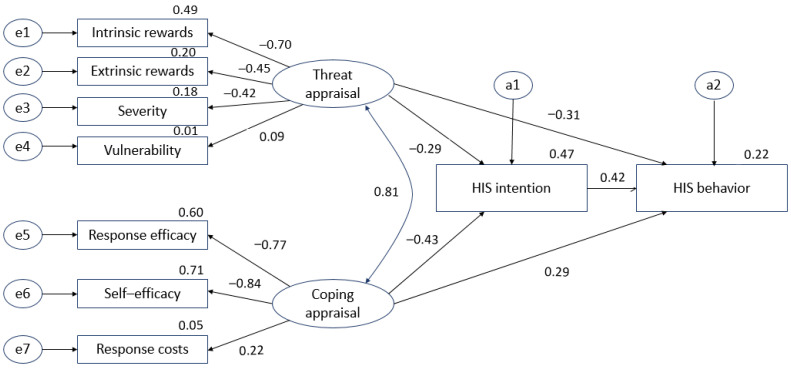
Initial hypothetical model. e1–e7: measurement error, a1–a2: structural error.

**Figure 3 ijerph-18-02084-f003:**
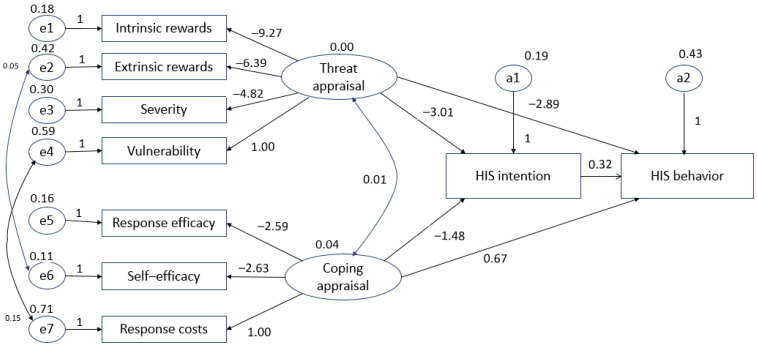
Revised hypothetical model.

**Table 1 ijerph-18-02084-t001:** Participants’ demographics (*N* = 252).

	Classification	*n*	%
Sex	Male	9	3.6
	Female	243	96.4
Age	≤24 years	12	4.8
	25–29 years	120	47.6
	30–34 years	73	29.0
	35–39 years	21	8.3
	≥40 years	26	10.3
Position	Staff	82	32.5
	Charge	161	63.9
	supervisor	9	3.6
Work experience	≤1 years	16	6.3
	1–4 years	92	36.5
	5–9 years	74	29.3
	≥10 years	70	27.8

**Table 2 ijerph-18-02084-t002:** Convergent validity (*N* = 252).

	Mean	SD	Cronbach’s α	AVE	CR
IR	3.6429	0.64049	0.947	0.930	0.976
ER	3.1518	0.72692	0.874	0.727	0.913
SEV	4.1437	0.60502	0.896	0.746	0.936
VUL	3.1958	0.79593	0.809	0.612	0.823
RE	3.9484	0.68370	0.758	0.731	0.844
SE	3.6984	0.61100	0.880	0.789	0.937
RC	2.7421	0.86752	0.772	0.508	0.755
HISI	3.9762	0.60082	0.856	0.775	0.932
HISB	4.0667	0.74833	0.869	0.562	0.865

SD, standard deviation; AVE, average variance extracted; CR, composite reliability; IR, intrinsic reward; ER, extrinsic reward; SEV, severity; VUL, vulnerability; RE, response efficacy; SE, self-efficacy; RC, response costs; HISI, healthcare information security intention; HISB, healthcare information security behavior.

**Table 3 ijerph-18-02084-t003:** Discriminant validity (*N* = 252).

	IR	ER	SEV	VU	RE	SE	RC	HISI	HISB
IR	0.930								
ER	0.179	0.727							
SEV	0.052	0.005	0.746						
VU	0.002	0.006	0.008	0.612					
RE	0.152	0.029	0.179	0.0004	0.731				
SE	0.197	0.126	0.080	0.002	0.322	0.789			
RC	0.027	0.0001	0.012	0.054	0.031	0.026	0.508		
HISI	0.199	0.044	0.131	0.004	0.240	0.320	0.048	0.775	
HISB	0.084	0.0003	0.062	0.037	0.030	0.229	0.052	0.167	0.562

**Table 4 ijerph-18-02084-t004:** Moderating effect of the length of a nurse’s career working in a hospital (*N* = 252).

Hypothesis	<5 Years				≥5 Years				
	Estimate	SE	CR	*p*	Estimate	SE	CR	*p*	Results
H6	−0.589	1.066	−0.553	0.580	35.607	184.936	0.193	0.847	
H7	−1.820	0.783	−2.323	*	0.458	1.750	0.262	0.793	Accepted
H8	−1.33	1.273	−1.04	0.917	83.424	438.604	0.190	0.849	
H9	0.320	0.654	0.489	0.625	4.027	4.668	0.863	0.388	
H10	0.600	−1.42	4.229	***	−0.421	1.044	−0.403	0.687	Accepted

SE, standard error. *** *p* < 0.001, * *p* < 0.05.

## Data Availability

The data presented in this study are available on request from the corresponding author.

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
