# Peer review of "Structural Model of the Healthcare Information Security Behavior of Nurses Applying Protection Motivation Theory"

_ijerph, 2021, doi:10.3390/ijerph18042084_

Round 1
Reviewer 1 Report
Thanks for the opportunity to read this manuscript. I have some comments before future process.
Research Hypotheses section needs improvement. It must to be written more clearly so that the reader has a better view of the analyzed issues.
The authors mentioned only one limitations. However, in my opinion, the study had more of them which should be stated.
The main problem with the article is clear style. In my opinion, professional Native Speaker support would be useful - preferably a researcher who would help to improve the language to make it more reader-friendly.
I highly recommend updating also some references, Please see and include following references:
http://dx.doi.org/10.2174/1874434602014010008
https://doi.org/10.1371/journal.pone.0244488
Also, please make sure your conclusions' section underscore the scientific value added of your paper, and the applicability of your findings.
Please revise your conclusion part into more details. Basically, you should enhance your contributions, limitations (as mentioned before), underscore the scientific value added of your paper, or the applicability of your findings and future study in this session
I look forward to read it again after improvements
Author Response
Response to Reviewer 1
ID: ijerph-1090834
<Thank you letter to reviewers>
We appreciate your critical review of our work and your suggestions for improving the quality of our manuscript. Based on the comments, we have provided point-by-point responses and have made the associated modifications to the manuscript.
Thank you in advance for your time and attention.
[Specific comments]
The main problem with the article is clear style. In my opinion, professional Native Speaker support would be useful - preferably a researcher who would help to improve the language to make it more reader-friendly.
[Response]
Thank you for your valuable comments. Based on your comments, the manuscript has been edited by a native speaker.
[Specific comments]
I highly recommend updating also some references, Please see and include following references:
http://dx.doi.org/10.2174/1874434602014010008[38]
https://doi.org/10.1371/journal.pone.0244488[39]
[Response]
This comment was very helpful for this paper, because you informed us of the latest literature. As you suggested, we have revised the manuscript as follows.
<page 10, lines 365-368>
Today, not only medical institutions but also major corporations and government organizations are facing the reality that they may be hacked [37], so information security policies should be implemented for each institution to ensure information security.
<page 10, lines 388-394>
Nurses are a very important group of healthcare professionals in medical institutions, and they must deal with disaster management [38], including HIS leakage. Nurses who have communication skills and knowledge on information utilization will be able to cope with emergency situations such as HIS leakage [38,39]. Major corporations and government organizations are actively protecting information through the use of new methods such as the FORGE system or MSN modeling [37,40]. Medical institutions may also develop new technologies for HIS or modify the previously mentioned methods to suit medical institutions
[Specific comments]
Also, please make sure your conclusions section underscore the scientific value added of your paper, and the applicability of your findings
[Response]
As you suggested, we have revised the manuscript as follow
<page 11, lines 423-433>
- Conclusions
As the possibility of healthcare information leakage increases, medical institutions are striving for information security. This study was conducted to explain the HIS behavior of nurses. The HIS behavior of nurses is very important because there are large numbers of nurses in medical institutions, and nurses are with patients 24 h a day. Prior studies only described the HIS intentions or behavior of nurses as parts of a variable, but this study explains the HIS intentions and behavior of nurses simultaneously by applying PMT. In this study, coping appraisal was found to influence HIS intentions, and HIS intentions were found to influence HIS behavior. Therefore, the results of this study indicate that in order to increase the adoption of HIS behavior, HIS intentions need to be improved, and for this, coping appraisal needs to be improved.
[Specific comments]
Please revise your conclusion part into more details. Basically, you should enhance your contributions, limitations (as mentioned before), underscore the scientific value added of your paper, or the applicability of your findings and future study in this session
[Response]
As you suggested, we have revised the manuscript as follows
<page 11, lines 409-422>
Generalization to all age groups has its limitations, but the data collected in this study can be best applied to nurses aged 25–29 years. This age group generally includes charge nurses, so the data can be used as basic information when creating policies for nurses in medical institutions. The other limitation of this study is that the only variable in the PMT was the description of nurses’ HIS behavior. It is necessary to measure HIS behavior according to various types of theories. However, the use of threat appraisal, coping appraisal, intention, and behavior in PMT is appropriate for explaining HIS. Prior studies only examined HIS intentions or behavior, but this study verified its methods using SEM to explain HIS intentions and behavior at the same time. Furthermore, prior studies only described HIS as a variable of PMT [6,10,15,28–30], but this study described HIS factors that interact with various variables in PMT. Our results indicate that nurses who have worked for more than five years become proficient in their work and have improved job satisfaction are more likely to exhibit HIS behavior than nurses who have worked for less than five years. Therefore, this study presented a suitable model for explaining and predicting the HIS behaviors of nurses.
Please see the attachment in the box.

Reviewer 2 Report
This is a very stimulating manuscript describing a novel model for explaining the healthcare information security (HIS) behaviour of nurses, based on protection motivation theory.
Major comments:
1) No explanations provided on the questionnaire supplied. Is it a validated questionnaire? Please, for further reproducibility, describe the statements of the questionnaires and its correspondence with the factors (IR, ER, SEV, VUEL, etc.).
2) More experiments are needed before ensuring this sentence in Discussion section: “In particular, it is suggested to target nurses with <5 years of work experience”. Why did you set the cutting point of working years in 5 years? Some well-founded reason? Did you try with other year ranges such as 2 or 4 working years? If yes, mention those experiment results. Maybe with 2 years the same results are found.
Minor comments:
1) In section “Participants and Procedures”. Why did you include nurses who worked at general hospitals with >= 300 beds for more than 1 moth at least? Any previous work references? Must be explained.
2) Why did you remove missing values from questionnaire? Have you considered alternative missing values technique in order to avoid the removal, such as interpolation?
3) The sample is imbalance in terms of the age feature. In this case, the highest percentage of recruited subjects is for ages ranged from 25-29, so the current model could not generalize for all of the people ages. Discuss or mention this as a limitation/future work. Future work should focus on balance the sample to be more robust in terms of generalization for all ages in South Korean nurses.
4) Consider any mention about nurses’ ages related with the final result about working years.
Author Response
Response to Reviewer 2
ID: ijerph-1090834
<Thank you letter to reviewers>
We appreciate your critical review of our work and your suggestions for improving the quality of our manuscript. Based on the comments, we have provided point-by-point responses and have made the associated modifications to the manuscript.
Thank you in advance for your time and attention.
[Specific comments]
Major comments:
1) No explanations provided on the questionnaire supplied. Is it a validated questionnaire? Please, for further reproducibility, describe the statements of the questionnaires and its correspondence with the factors (IR, ER, SEV, VUEL, etc.).
[Response]
Thank you for your valuable, detailed comments. In this paper, we considered endogenous variables, mediating variables, exogenous variables, and subfactors of exogenous variables. Based on your comments, we have revised the manuscript as follows:
<page 4, lines 175-232>
3.3. Measures
To measure the concept of nurses’ HIS behavior, we used the PMT variables proposed by Rogers and Price-Dunn [17]. Threat appraisal and coping appraisal were set as independent variables, and HIS intention and HIS behavior were set as dependent variables [16,17]. Intrinsic rewards, extrinsic rewards, severity, and vulnerability were selected as subfactors of threat appraisal [17,25]. Response efficacy, self-efficacy, and response costs were selected as subfactors of coping appraisal [17,27]. Career length was set as a moderate variable [10].
In this study, HIS intention was measured by the tools used by Kim [32]. The measurement tool involved five questions, each answered on a five-point Likert scale ranging from 1 (“strongly disagree”) to 5 (“strongly agree”): The higher the score, the higher the intention for HIS. In a study by Kim [32], the value of Cronbach’s α was 0.890, and it was 0.856 in this study.
In this study, HIS behavior was measured by the tools used by Kim [32]. The measurement tool included 10 questions, each answered on a five-point Likert scale ranging from 1 (“strongly disagree”) to 5 (“strongly agree”): The higher the score, the more strongly the HIS behavior was practiced. In a study by Kim [32], the value of Cronbach’s α was 0.765, and it was 0.869 in this study.
Intrinsic rewards describe one’s satisfaction or sense of accomplishment [17,25]. In this study, intrinsic rewards were measured by the tools used by Kim et al. [33]. The measurement tool involved three questions, each answered on a five-point Likert scale ranging from 1 (“strongly disagree”) to 5 (“strongly agree”): The higher the score, the higher the intrinsic rewards for HIS. In a study by Kim et al. [33], the value of Cronbach’s α was 0.954, and it was 0.947 in this study.
Extrinsic rewards include social consensus, peer influence, and education [17,25]. In this study, extrinsic rewards were measured by the tools used by Kim et al. [33]. The measurement tool involved four questions, each answered on a five-point Likert scale ranging from 1 (“strongly disagree”) to 5 (“strongly agree”): The higher the score, the higher the extrinsic rewards for HIS. In a study by Kim et al. [33], the value of Cronbach’s α was 0.931, and it was 0.874 in this study.
Severity is the extent to which a hazard is fatal if it occurs [17,25]. In this study, severity was measured by the tools used by Jung [34]. The measurement tool involved six questions, each answered on a five-point Likert scale ranging from 1 (“strongly disagree”) to 5 (“strongly agree”): The higher the score, the higher the severity related to HIS. In a study by Jung [34], the value of CSRI was 0.942, and the value of Cronbach’s α was 0.896 in this study.
Vulnerability is the possibility that a hazard will actually occur [17,27]. In this study, vulnerability was measured by the tools used by Jung [34]. The measurement tool involved six questions, each answered on a five-point Likert scale ranging from 1 (“strongly disagree”) to 5 (“strongly agree”): The higher the score, the higher the vulnerability related to HIS. In a study by in Jung [34], the value of CSRI was 0.945, and Cronbach’s α was 0.809 in this study.
Response efficacy is whether the proposed policy has the effect of preventing the hazard policy [17,25,27]. In this study, response efficacy was measured by the tools used by Son [35]. The measurement tool involved four questions, each answered on a five-point Likert scale ranging from 1 (“strongly disagree”) to 5 (“strongly agree”): The higher the score, the higher the response efficacy related to HIS. In a study by Son [35], the value of Cronbach’s α was 0.877, and it was 0.758 in this study.
Self-efficacy involves the self-assessment of whether one can carry out the proposed policy [17,25]. In this study, self-efficacy was measured by the tools used by Son [35]. The measurement tool involved four questions, each answered on a five-point Likert scale ranging from 1 (“strongly disagree”) to 5 (“strongly agree”): The higher the score, the higher the self-efficacy related to HIS. In a study by Son [35], the value of Cronbach’s α was 0.929, and it was 0.880 in this study.
Response costs represent obstacles, such as time and hassle, that make it difficult to actually conduct a process [17,27]. In this study, response costs were measured by the tools used by Son [35]. The measurement tool involved three questions, each answered on a five-point Likert scale ranging from 1 (“strongly disagree”) to 5 (“strongly agree”): The higher the score, the higher the response costs related to HIS. In a study by Son [35], the value of Cronbach’s α was 0.795, and it was 0.772 in this study. Nurses’ careers were classified as less than or more than five years [9,10,15].
[Specific comments]
Major comments:
2) More experiments are needed before ensuring this sentence in Discussion section: “In particular, it is suggested to target nurses with <5 years of work experience”. Why did you set the cutting point of working years in 5 years? Some well-founded reason? Did you try with other year ranges such as 2 or 4 working years? If yes, mention those experiment results. Maybe with 2 years the same results are found.
[Response]
We added literature that found different results for nurses with more and less than five years of work experience. In addition, a questionnaire with a small working range was prepared to see the moderate effects in the SEM, and this was done to check whether there were any differences in the general characteristics of the nurses depending on their work experience.
As you suggested, we have revised the manuscript as follow
<page 2, lines 64-79>
Nurses are hesitate to HIS behavior during the transfer of a patient to another department for examination or when performing nursing services such as medication or wound dressing, at an exact time for patient [10]. Nursing services, such as medication or wound dressing, are performed in the patient’s hospital room, and HIS behavior is performed at the nurses’ station. These tasks should be done in different, separated spaces to hesitate to HIS behavior [10,14]. In unpredictable and urgent emergencies, such as for those requiring CPR(CardioPulmonary Resuscitation), HIS behavior can be delayed and missed [10]. Additionally, nurses with lengthier careers are statistically more receptive to and proficient in HIS [15]. Nurses who have worked for approximately five years have increased job satisfaction and become proficient in their work [9, 10]. In addition, they become better at dealing with crisis situations while conducting their nursing services and often become charge nurses on the ward, meaning that they are in charge of making decisions [9,15]. The Korean Health Industry Promotion Agency conducted a fact-finding survey in which nurses were divided into groups of least five years based on their length of time in the professions with less consideration of their careers [6,10,15].
[Specific comments]
Minor comments:
1) In section “Participants and Procedures”. Why did you include nurses who worked at general hospitals with >= 300 beds for more than 1 moth at least? Any previous work references? Must be explained.
[Response]
Legally, in South Korea, hospitals with more than 300 beds have nine or more medical personnel, including those working in internal medicine, surgery, pediatrics, radiology, anesthesia and pain, or pathology, psychiatry, and dentistry, with full-time specialists in each course of care.
<page 4, lines 165-169>
Hospitals with more than 300 beds operate more medical departments, so OCS, EMR, and PACS are equipped to process healthcare information electronically, making it appropriate to investigate nurses’ HIS. Nurses with less than one month of work experience were excluded, as they do not yet perform HIS behavior independently because they perform all duties under supervision in one ward during the training period.
[Specific comments]
Minor comments:
2) Why did you remove missing values from questionnaire? Have you considered alternative missing values technique in order to avoid the removal, such as interpolation?
[Response]
Based on your comments, we added the following statement to the manuscript:
<page 6, lines 246-247>
After removing questionnaires containing incomplete or missing responses to more than 30% of the questions
[Specific comments]
Minor comments:
3) The sample is imbalance in terms of the age feature. In this case, the highest percentage of recruited subjects is for ages ranged from 25-29, so the current model could not generalize for all of the people ages. Discuss or mention this as a limitation/future work. Future work should focus on balance the sample to be more robust in terms of generalization for all ages in South Korean nurses.
[Response]
Based on your comments, we added the following statement to the manuscript:
<page 11, lines 409-412>
Generalization to all age groups has its limitations, but the data collected in this study can be best applied to nurses aged 25–29 years. This age group generally includes charge nurses, so the data can be used as basic information when creating policies for nurses in medical institutions.
<pages 10-11, lines 404-406>
This result shows that an increased career length is associated with repeated educated and thus improved awareness of the need for, and effect of, using a security system.
[Specific comments]
Minor comments:
4) Consider any mention about nurses’ ages related with the final result about working years.
[Response]
Thank you for your valuable comments. Based on your comments, we added the following statement to the manuscript:
<page 10, lines 400-401>This result shows that the greater the work experience, the more HIS behavior is displayed.
Please see the attachment in the box.

Reviewer 3 Report
The authors propose a study about nurses' healthcare information security by providing 252 questionnaires scored using a five point Likert scale.
The proposed study is interesting but there are some points that the authors should better discuss.
The authors should be better described the novelties of their study with respect to existing ones. In particular, the author should discuss limitation and cons of the examined approaches. Furthermore, the authors should provide more details and discussion about the obtained results. The Discussion section also needs to be improved by analyzing the outcome of evaluation section.
I suggest to further analyze more recent approaches about the examined topics. In particular, I suggest the following papers to further investigate deception issues and multimedia analysis in Healthcare Information System (HIS):
1) FORGE: a fake online repository generation engine for cyber deception. IEEE Transactions on Dependable and Secure Computing.
2) Multimedia summarization using social media content. Multimedia Tools and Applications, 77(14), 17803-17827.
Finally, I suggest to perform a linguistic revision.
Author Response
Response to Reviewer 3
ID: ijerph-1090834
<Thank you letter to reviewers>
We appreciate your critical review of our work and your suggestions for improving the quality of our manuscript. Based on the comments, we have provided point-by-point responses and have made the associated modifications to the manuscript.
Thank you in advance for your time and attention.
[Specific comments]
The authors should be better described the novelties of their study with respect to existing ones. In particular, the author should discuss limitation and cons of the examined approaches. Furthermore, the authors should provide more details and discussion about the obtained results. The Discussion section also needs to be improved by analyzing the outcome of evaluation section.
[Response]
Thank you for your valuable comments. Based on your comments, we added the following statement to the manuscript:
<page 11, lines 409-332>
Generalization to all age groups has its limitations, but the data collected in this study can be best applied to nurses aged 25–29 years. This age group generally includes charge nurses, so the data can be used as basic information when creating policies for nurses in medical institutions. The other limitation of this study is that the only variable in the PMT was the description of nurses’ HIS behavior. It is necessary to measure HIS behavior according to various types of theories. However, the use of threat appraisal, coping appraisal, intention, and behavior in PMT is appropriate for explaining HIS. Prior studies only examined HIS intentions or behavior, but this study verified its methods using SEM to explain HIS intentions and behavior at the same time. Furthermore, prior studies only described HIS as a variable of PMT [6,10,15,28–30], but this study described HIS factors that interact with various variables in PMT. Our results indicate that nurses who have worked for more than five years become proficient in their work and have improved job satisfaction are more likely to exhibit HIS behavior than nurses who have worked for less than five years. Therefore, this study presented a suitable model for explaining and predicting the HIS behavior of nurses.
- Conclusions
As the possibility of healthcare information leakage increases, medical institutions are striving for information security. This study was conducted to explain the HIS behavior of nurses. The HIS behavior of nurses is very important because there are large numbers of nurses in medical institutions, and nurses are with patients 24 h a day. Prior studies only described the HIS intentions or behavior of nurses as parts of a variable, but this study explains the HIS intentions and behavior of nurses simultaneously by applying PMT. In this study, coping appraisal was found to influence HIS intentions, and HIS intentions were found to influence HIS behavior. Therefore, the results of this study indicate that in order to increase the adoption of HIS behavior, HIS intentions need to be improved, and for this, coping appraisal needs to be improved.
[Specific comments]
I suggest to further analyze more recent approaches about the examined topics. In particular, I suggest the following papers to further investigate deception issues and multimedia analysis in Healthcare Information System (HIS):
1) FORGE: a fake online repository generation engine for cyber deception. IEEE Transactions on Dependable and Secure Computing [37]
2) Multimedia summarization using social media content. Multimedia Tools and Applications, 77(14), 17803-17827. [40]
[Response]
This comment was very helpful, because you informed us of the latest literature. Based on your comments, we added the following statement to the manuscript:
<page 10, lines 388-394>
Nurses are a very important group of healthcare professionals in medical institutions, and they must deal with disaster management [38], including HIS leakage. Nurses who have communication skills and knowledge on information utilization will be able to cope with emergency situations such as HIS leakage [38,39]. Major corporations and government organizations are actively protecting information through the use of new methods such as the FORGE system or MSN modeling [37,40]. Medical institutions may also develop new technologies for HIS or modify the previously mentioned methods to suit medical institutions.
[Specific comments]
Finally, I suggest to perform a linguistic revision.
[Response]
Thank you for your comment. Our manuscript has been edited.
Please see the attachment in the box.

Reviewer 4 Report
This study examines the health information security behavior among nurses in South Korea. Overall, this is a well-written manuscript with proper theory and methods. Here are some suggestions to improve the manuscript further.
Keywords:
- The authors may want to include South Korea as a keyword.
Introduction:
- Page 2, the sentence beginning with “However, physical or environmental obstacles”: authors can give examples of what these obstacles are.
- Page 2: the authors could explain why they included nurses only and if there are any differences in health information security behaviors among doctors, nurses, and staff members at the hospital found in the previous studies.
Research Hypotheses:
- This section is written clearly, but I think the authors need to relocate the subsection, 3.1.1. Protection Motivation Theory before their hypotheses. The readers who are not familiar with the Protection Motivation Theory need this theoretical background before reading hypotheses.
- Hypotheses 6-10 are presented on page 7, but they should have appeared in this section with the supporting findings from the previous studies.
Research Methodology:
- I think this section needs the most revisions. It is unclear whether the consent was written consent or verbal consent. Also, was there any financial reward to participate in this study?
- The authors should indicate the sampling method.
- Measures: The authors need to give a detailed description of the measures of their dependent variables (HIS behaviors and intentions), independent variables (threat appraisal and coping appraisal), and a moderation variable (the nurses’ careers). The variables which were included in Tables 2, 3, and figures 2 and 3 should be explained further. For example, how did the intrinsic rewards measured with the Likert scale? What was the original question?
Conclusions:
- Please do not end the study with its limitation. Instead, emphasize the significance of the study, implications of the study, and suggestions for future studies.
Author Response
Response to Reviewer 4
ID: ijerph-1090834
<Thank you letter to reviewers>
We appreciate your critical review of our work and your suggestions for improving the quality of our manuscript. Based on the comments, we have provided point-by-point responses and have made the associated modifications to the manuscript.
Thank you in advance for your time and attention.
[Specific comments]
Keywords:
- The authors may want to include South Korea as a keyword.
[Response]
Based on your comments, we added the following keyword to the manuscript: South Korea
[Specific comments]
Introduction:- Page 2, the sentence beginning with “However, physical or environmental obstacles”: authors can give examples of what these obstacles are.
[Response]
As you commented, we have included the following text:
<page 2, lines 65-79>
Nurses are hesitate to HIS behavior during the transfer of a patient to another department for examination or when performing nursing services such as medication or wound dressing, at an exact time for patient [10]. Nursing services, such as medication or wound dressing, are performed in the patient’s hospital room, and HIS behavior is performed at the nurses’ station. These tasks should be done in different, separated spaces to hesitate to HIS behavior [10,14]. In unpredictable and urgent emergencies, such as for those requiring CPR(CardioPulmonary Resuscitation), HIS behavior can be delayed and missed [10]. Additionally, nurses with lengthier careers are statistically more receptive to and proficient in HIS [15]. Nurses who have worked for approximately five years have increased job satisfaction and become proficient in their work [9, 10]. In addition, they become better at dealing with crisis situations while conducting their nursing services and often become charge nurses on the ward, meaning that they are in charge of making decisions [9,15]. The Korean Health Industry Promotion Agency conducted a fact-finding survey in which nurses were divided into groups of least five years based on their length of time in the professions with less consideration of their careers [6,10,15].
[Specific comments]
Introduction:
- Page 2: the authors could explain why they included nurses only and if there are any differences in health information security behavior among doctors, nurses, and staff members at the hospital found in the previous studies.
[Response]
All medical personnel are important, but in this study, we investigated the HIS behavior of nurses who reside with patients 24 h a day and occupy a large part of the workforce in the hospital.
As you suggested, we have included the following statement:
<page 2, lines 52-53>
In medical institutions, nurses operate within large workplaces and are in contact with patients for 24 h per day [1].
[Specific comments]
Research Hypotheses:- This section is written clearly, but I think the authors need to relocate the subsection, 3.1.1. Protection Motivation Theory before their hypotheses. The readers who are not familiar with the Protection Motivation Theory need this theoretical background before reading hypotheses.
[Response]
Based on your comments, we have changed the order as follow
2.1. Theoretical Foundation
2.1.1. Protection Motivation Theory
[Specific comments]
Research Hypotheses:
- Hypotheses 6-10 are presented on page 7, but they should have appeared in this section with the supporting findings from the previous studies.
[Response]
Thank you for your valuable comments. Based on your comments, we have added the rationale for hypotheses 610 as follow
<page 9, lines 315-318>
The greater a nurse’s experience, the more significant an impact HIS has on behavior [15]. In particular, nurses who have worked for more than five years have been found to be statistically significantly more likely to carry out HIS behavior than nurses who have worked for less than five years [9].
<page 9, lines 322-323>
......to identify the moderating effect of having more than five years of experience:
[Specific comments]
Research Methodology:
- I think this section needs the most revisions. It is unclear whether the consent was written consent or verbal consent. Also, was there any financial reward to participate in this study?
[Response]
All participants signed the agreement after receiving an explanation. We received written consent forms. Based on your comments, we have revised the manuscript as follow
<page 4, lines 169-170>
After obtaining approval from the nursing department, a public notice for the recruitment of eligible persons was posted.
<page 4, lines 153-157>
It was explained that the anonymity of the subjects was guaranteed and that the findings would not be used for any purpose other than research in the future. Nurses participated in the survey after being informed that it would take approximately 20 min. After explanation, all of them signed the consent form. The nurses who participated in the survey were given a small travel wash kit to thank them.
[Specific comments]
Research Methodology:
- The authors should indicate the sampling method.
[Response]
Based on your comments, we have included the following text:
<page 4, line 160>
in Seoul and Gyeonggi-do, the largest area near Seoul, South Korea
<page 4, lines 164-169>
Hospitals with more than 300 beds operate more medical departments, so OCS, EMR, and PACS are equipped to process healthcare information electronically, making it appropriate to investigate nurses’ HIS. Nurses with less than one month of work experience were excluded, as they do not yet perform HIS behavior independently because they perform all duties under supervision in one ward during the training period.
[Specific comments]
Research Methodology:
- Measures: The authors need to give a detailed description of the measures of their dependent variables (HIS behavior and intentions), independent variables (threat appraisal and coping appraisal), and a moderation variable (the nurses’ careers). The variables which were included in Tables 2, 3, and figures 2 and 3 should be explained further. For example, how did the intrinsic rewards measured with the Likert scale? What was the original question?
[Response]
Based on your comments, we have included the following text:
<pages 4-6 lines 174-232>
3.3. Measures
To measure the concept of nurses’ HIS behavior, we used the PMT variables proposed by Rogers and Price-Dunn [17]. Threat appraisal and coping appraisal were set as independent variables, and HIS intention and HIS behavior were set as dependent variables [16,17]. Intrinsic rewards, extrinsic rewards, severity, and vulnerability were selected as subfactors of threat appraisal [17,25]. Response efficacy, self-efficacy, and response costs were selected as subfactors of coping appraisal [17,27]. Career length was set as a moderate variable [10].
In this study, HIS intention was measured by the tools used by Kim [32]. The measurement tool involved five questions, each answered on a five-point Likert scale ranging from 1 (“strongly disagree”) to 5 (“strongly agree”): The higher the score, the higher the intention for HIS. In a study by Kim [32], the value of Cronbach’s α was 0.890, and it was 0.856 in this study.
In this study, HIS behavior was measured by the tools used by Kim [32]. The measurement tool included 10 questions, each answered on a five-point Likert scale ranging from 1 (“strongly disagree”) to 5 (“strongly agree”): The higher the score, the more strongly the HIS behavior was practiced. In a study by Kim [32], the value of Cronbach’s α was 0.765, and it was 0.869 in this study.
Intrinsic rewards describe one’s satisfaction or sense of accomplishment [17,25]. In this study, intrinsic rewards were measured by the tools used by Kim et al. [33]. The measurement tool involved three questions, each answered on a five-point Likert scale ranging from 1 (“strongly disagree”) to 5 (“strongly agree”): The higher the score, the higher the intrinsic rewards for HIS. In a study by Kim et al. [33], the value of Cronbach’s α was 0.954, and it was 0.947 in this study.
Extrinsic rewards include social consensus, peer influence, and education [17,25]. In this study, extrinsic rewards were measured by the tools used by Kim et al. [33]. The measurement tool involved four questions, each answered on a five-point Likert scale ranging from 1 (“strongly disagree”) to 5 (“strongly agree”): The higher the score, the higher the extrinsic rewards for HIS. In a study by Kim et al. [33], the value of Cronbach’s α was 0.931, and it was 0.874 in this study.
Severity is the extent to which a hazard is fatal if it occurs [17,25]. In this study, severity was measured by the tools used by Jung [34]. The measurement tool involved six questions, each answered on a five-point Likert scale ranging from 1 (“strongly disagree”) to 5 (“strongly agree”): The higher the score, the higher the severity related to HIS. In a study by Jung [34], the value of CSRI was 0.942, and the value of Cronbach’s α was 0.896 in this study.
Vulnerability is the possibility that a hazard will actually occur [17,27]. In this study, vulnerability was measured by the tools used by Jung [34]. The measurement tool involved six questions, each answered on a five-point Likert scale ranging from 1 (“strongly disagree”) to 5 (“strongly agree”): The higher the score, the higher the vulnerability related to HIS. In a study by in Jung [34], the value of CSRI was 0.945, and Cronbach’s α was 0.809 in this study.
Response efficacy is whether the proposed policy has the effect of preventing the hazard policy [17,25,27]. In this study, response efficacy was measured by the tools used by Son [35]. The measurement tool involved four questions, each answered on a five-point Likert scale ranging from 1 (“strongly disagree”) to 5 (“strongly agree”): The higher the score, the higher the response efficacy related to HIS. In a study by Son [35], the value of Cronbach’s α was 0.877, and it was 0.758 in this study.
Self-efficacy involves the self-assessment of whether one can carry out the proposed policy [17,25]. In this study, self-efficacy was measured by the tools used by Son [35]. The measurement tool involved four questions, each answered on a five-point Likert scale ranging from 1 (“strongly disagree”) to 5 (“strongly agree”): The higher the score, the higher the self-efficacy related to HIS. In a study by Son [35], the value of Cronbach’s α was 0.929, and it was 0.880 in this study.
Response costs represent obstacles, such as time and hassle, that make it difficult to actually conduct a process [17,27]. In this study, response costs were measured by the tools used by Son [35]. The measurement tool involved three questions, each answered on a five-point Likert scale ranging from 1 (“strongly disagree”) to 5 (“strongly agree”): The higher the score, the higher the response costs related to HIS. In a study by Son [35], the value of Cronbach’s α was 0.795, and it was 0.772 in this study. Nurses’ careers were classified as less than or more than five years [9,10,15].
[Specific comments]
Conclusions:
- Please do not end the study with its limitation. Instead, emphasize the significance of the study, implications of the study, and suggestions for future studies.
[Response]
Thank you for your valuable comments. Based on your comments, we have revised the manuscript as follow
<page 11, lines 409-433>
Generalization to all age groups has its limitations, but the data collected in this study can be best applied to nurses aged 25–29 years. This age group generally includes charge nurses, so the data can be used as basic information when creating policies for nurses in medical institutions. The other limitation of this study is that the only variable in the PMT was the description of nurses’ HIS behavior. It is necessary to measure HIS behavior according to various types of theories. However, the use of threat appraisal, coping appraisal, intention, and behavior in PMT is appropriate for explaining HIS. Prior studies only examined HIS intentions or behavior, but this study verified its methods using SEM to explain HIS intentions and behavior at the same time. Furthermore, prior studies only described HIS as a variable of PMT [6,10,15,28–30], but this study described HIS factors that interact with various variables in PMT. Our results indicate that nurses who have worked for more than five years become proficient in their work and have improved job satisfaction are more likely to exhibit HIS behavior than nurses who have worked for less than five years. Therefore, this study presented a suitable model for explaining and predicting the HIS behavior of nurses.
- Conclusions
As the possibility of healthcare information leakage increases, medical institutions are striving for information security. This study was conducted to explain the HIS behavior of nurses. The HIS behavior of nurses is very important because there are large numbers of nurses in medical institutions, and nurses are with patients 24 h a day. Prior studies only described the HIS intentions or behavior of nurses as parts of a variable, but this study explains the HIS intentions and behavior of nurses simultaneously by applying PMT. In this study, coping appraisal was found to influence HIS intentions, and HIS intentions were found to influence HIS behavior. Therefore, the results of this study indicate that in order to increase the adoption of HIS behavior, HIS intentions need to be improved, and for this, coping appraisal needs to be improved.
Please see the attachment in the box.

Round 2
Reviewer 1 Report
The authors have addressed each of my comment. The explanation provided is clear, job well done.
Reviewer 2 Report
Congratulations for your revised work. All that effort has contributed to improve the quality of the paper.
Reviewer 3 Report
I think that the authors have addressed all my concerns